Corrected: Author correction

# Limitation of Fermi level shifts by polaron defect states in hematite photoelectrodes

Christian Lohaus[1], Andreas Klein [1] & Wolfram Jaegermann[1]

The optical band gap is a major selection criterion for an absorber in photocatalytic water splitting. Due to its ideal value hematite has been intensively investigated without reaching the expectation, yet. In this work, the Fermi level positions in hematite due to doping and contact formation are investigated. An upper boundary for the Fermi level position at 1.8 eV above the valence band maximum due to the formation of polarons is identified. This results in a different concept of the effective band gap for hematite which we believe is transferable to any material with competing polaron formation after optical excitation: the optical band gap of 2.2 eV deviates from an effective electronic band gap of 1.75 eV. The polaron state acts as a limit in (quasi-)Fermi level shift, restricting the potential of charge transfer reactions. Additionally, it has led to an incorrect determination of the band edge positions of hematite in electrochemical contacts.

[1] Surface Science Division, Materials Science Department, TU, Darmstadt, Germany. Correspondence and requests for materials should be addressed to W.J. (email: jaegermann@surface.tu-darmstadt.de)

The photoelectrochemical splitting of water into $H_2$ and $O_2$ as illustrated in Fig. 1a is considered as a promising route to generate renewable fuels since the early experiments of Fujishima and Honda[1]. From theoretical considerations it can be deduced that a bandgap of the semiconductor of $E_g \geq 2$ eV is needed in order to generate a high enough photovoltage by splitting of the quasi-Fermi levels[2–5]. The search for materials that meet this requirement is based e.g. on experimental and computational high-throughput methods[6].

Hematite ($\alpha$-$Fe_2O_3$) with its energy gap of 2.2 eV has been considered as an ideal candidate as it is non-hazardous, easy to prepare, abundant, and stable in aqueous environment[5,7]. However, achieved conversion efficiencies are lacking far behind the calculated theoretical level of 16.8%[8,9]. In particular the photovoltages are far below the expected values[4,10,11]. The discrepancy between theoretical and achieved efficiencies has been assigned to poor charge separation due to low mobility of electrons[9,12], a non-suitable position of the $Fe_2O_3$ conduction band with respect to the hydrogen evolution potential[2,13], and to Fermi level pinning at surface states[14,15]. To overcome these limitations, doping, nanostructuring, surface and light-harvesting optimization, and addition of co-catalysts have been suggested[14,16–20]. But none of these approaches could bring the conversion efficiency of $Fe_2O_3$ close to the theoretical limit.

In a classical semiconductor, such as Si or GaAs, the Fermi energy can easily be manipulated by doping and typically it can be varied throughout the whole energy gap and even beyond. In such cases the optical and electronical gap are identical (neglecting exciton formation). In organic semiconductors variations of the effective bandgaps due to polaron formation and/or excitonic effects are well accepted and considered in the performance of photochemical energy converter[21].

In oxidic semiconductors there is another fundamental thermodynamic limit for the variation of the Fermi energy. This is known as self-compensation, which is related to the formation of ionic defects in response to a variation of the Fermi energy[22–24]. Any further doping of a material will in this case not result in the generation of additional free electrons or holes, but in the generation of ionic defects, such as oxygen vacancies or metal interstitial. These phenomena are well described by defect chemistry or Fermi energy-dependent defect formation enthalpies[22,24]. These mechanisms result in an upper and lower limit for the Fermi energy variation in a material as e.g. for some oxides and sulfides[22,23]. Another mechanism limiting the variation of the Fermi energy in a material is related to the energy level of the dopant. The Fermi level cannot be raised above the energy level of a donor or below the level of an acceptor, as the dopants will become electrically neutral. The consideration of the change of the oxidation state of the host atoms defining a well-defined limitation value (energy state of the redox level) is a fundamentally valid and in the past already well studied bulk property of semiconductors with localized electronic states but an often overlooked mechanism for the limitation of Fermi energy shifts in a material. The Fermi energies at which the host atoms change their oxidation state can be considered as defect energy levels, with the important difference that the potential defect concentration is the same as the number of atoms in the material. Moreover, the defect is only present if the Fermi energy reaches the corresponding level. In contrast to self-compensation considered for stoichiometric charge compensation, the change of oxidation state for polaron formation does not require the formation of lattice defects, which is associated with atom movement. While self-compensation can therefore be kinetically hindered, no such limitation is expected for changing an oxidation state in polarons.

As a consequence of these limitations of the Fermi level position by doping, contacts, or illumination there is a fundamental limitation expected in the energetic position available in photovoltaic and photoelectrochemical converters. Whereas polaron-induced limitation for the charge carrier mobility is well aware in the scientific community, the limitations in the available potentials for charge transfer have hardly been taken under consideration so far.

In order to evaluate the accessible range of the Fermi energy in oxidic semiconductors with localized electronic states we have chosen hematite as prototype case. We performed photoelectron spectroscopy (PES) of differently doped $Fe_2O_3$ thin films prepared in situ by reactive magnetron sputtering. Different doping types and levels were achieved by co-sputtering using Mg, Si, and Zr targets, which are typical acceptor and donor dopants of $Fe_2O_3$[25–27]. Details of the deposition parameters and the achieved doping concentrations can be found in the Methods and Supplementary Methods.

In this work we demonstrate using PES of differently doped $Fe_2O_3$ thin films and their interface formation that the low efficiencies of $Fe_2O_3$ water-splitting devices with too low photovoltages is caused by the inability of the material to establish a sufficient Fermi level splitting under illumination. The limitation is caused by the reduction of Fe from $Fe^{3+}$ to $Fe^{2+}$, corresponding to the formation of electron polarons, if the Fermi energy is raised to a value ~0.5 eV below the conduction band minimum. We are convinced that the observation corresponds to a fundamental limitation of the electronically usable energy gaps in transition-metal compounds with strongly localized metal $d^n$ electron states. Instead of the optical bandgap, which corresponds to the initial excitation of electrons from occupied valence band to unoccupied conduction band states, a significantly smaller polaron gap becomes relevant if slow charge transfer processes are involved. The polaron gap is determined by the Fermi energy, at which the host atoms of a material change their oxidation state. We further demonstrate that the optical conduction band minimum is correct, as CBM energy of $Fe_2O_3$ is initially energetically above the hydrogen evolution potential as needed for an ideal photocatalyst (Fig. 1a) but that the poor hydrogen evolution is caused by the $Fe^{3+/2+}$ polaron level, which is below the $H_2$ evolution energy.

## Results

**Fermi level positions in doped hematite.** The crystallographic structure, valence band features as well as the optical energy gap of the prepared films correspond well with literature[28–30]. The latter two are displayed in Fig. 1b. The upper left inset shows the dependency of the absorption coefficient of a $Fe_2O_3$ thin films on the photon energy[28]. The Fermi energies of the different films, which are derived from the onset of the valence band emissions (also shown in Fig. 1b), which are displayed in Fig. 2a, are distinctly different for undoped (blue circles), Zr-doped (red diamonds), Si-doped (yellow pentagons), and Mg-doped (green stars) films. The variation of Fermi levels for a given doping is attributed to varying dopant and defect concentrations depending on the deposition conditions[31–33]. As expected, lowest Fermi energies are observed for acceptor (Mg) doped samples and higher Fermi energies for donor (Si and Zr) doping. The observation that Zr-doping results in higher Fermi energies as compared to Si-doping is attributed to the reduction of Si donors to $Si^{3+}$. The highest Fermi energy observed for Si-doped films of $E_F - E_{VBM} = 1.34$ eV corresponds well with a calculated $Si^{4+/3+}$ charge transition at $E_F - E_{VBM} = 1.41$ eV[26]. For Zr only a reduction to $Zr^0$ is expected, which will occur at much higher Fermi energies[34]. Please note that the same Fermi level positions can be found in Supplementary Figure 9 in more detail. Optical spectra in Supplementary Figure 10 for the differently

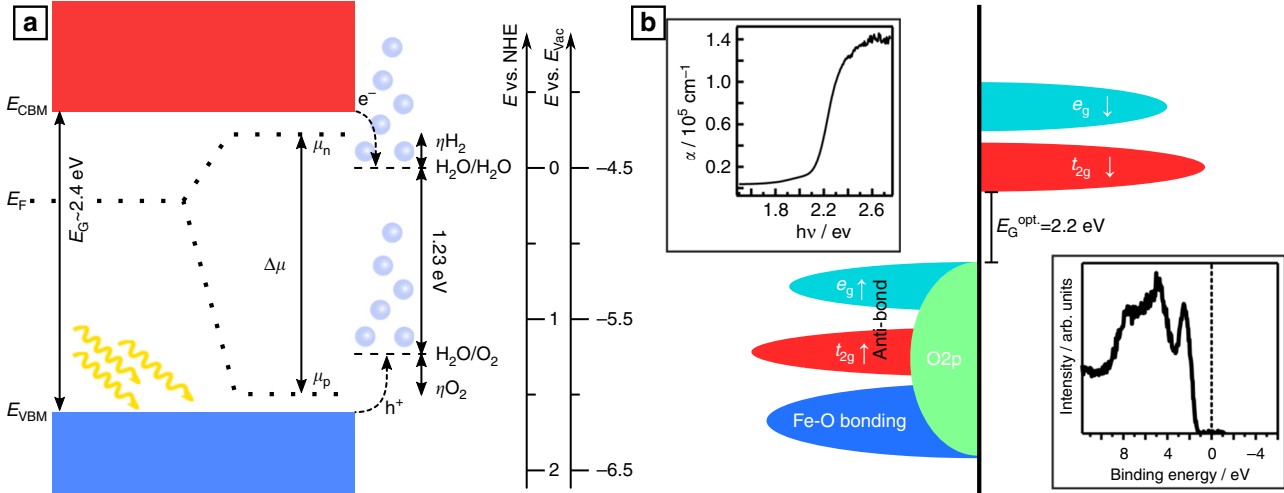

**Fig. 1** The idealized water-splitting process and the electronic structure of hematite. **a** The light-driven water-splitting process on an ideal material. Water splitting is possible once the splitting of the quasi-Fermi level $\Delta\mu$ is >1.23 eV plus additional overvoltages $\eta$ (adding up to about 1.8 eV). For this purpose the Fermi level must be able to shift throughout the bandgap. **b** Schematic representation of the electronic structure of hematite. The insets show a representative optical absorption coefficient and photoemission valence band as measured on our samples

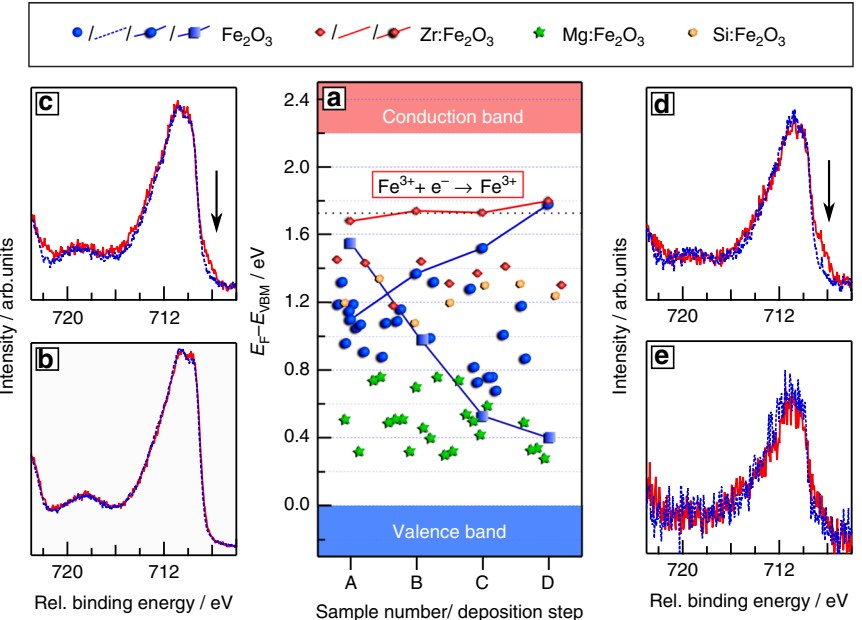

**Fig. 2** Achievable Fermi level positions in hematite. In **a** the measured Fermi level positions for nominally undoped (blue circles) and doped thin films with different dopants (red diamonds: Zr, green stars: Mg, and yellow pentagons: Si) are presented. In addition the Fermi level position for interface experiments toward ITO on an undoped (blue line and circles) and a Zr-doped (red line and diamonds) thin films and toward NiO on an undoped thin film (blue line and squares) are shown. The comparison of the corresponding Fe2p spectra at the different stages of the experiments are given in the **b**–**e** (corresponding to **b**: 0 s, **c**: 5 s, **d**: 10 s, and **e**: 20 s of accumulated ITO deposition time). The presence of the $Fe^{2+}$ signal corresponds to the Fermi level position

doped samples revealed no drastic changes of the optical properties.

The lowest and highest Fermi energies for differently doped samples deposited with varying substrate temperature and oxygen pressures are $E_F - E_{VBM} = 0.25$ and 1.68 eV, respectively. The overall variation of 1.4 eV is comparable to that observed for other oxides such as $SrTiO_3$ and $PbZrO_3$[35]. For all these samples the oxidation state of iron is always +III, as indicated by the satellite structure of the $Fe2p_{3/2}$ core levels shown in Fig. 2b[28,36]. Further $Fe2p_{3/2}$ core-level spectra of differently doped samples, including a wide variation of Fermi energy are displayed in

Supplementary Figure 3. All spectra exhibit identical line shapes and the core-level binding energies are shifted in parallel to the valence band spectra, confirming that the shifts are caused by different Fermi energy positions. The substantial variation of Fermi energy with doping and deposition conditions rules out surface Fermi level pinning, which has been claimed to be responsible for the too low photovoltages of hematite[14,15].

**Fermi level variations in hematite by interface formation.** A variation of the Fermi energy at the surface can also be obtained

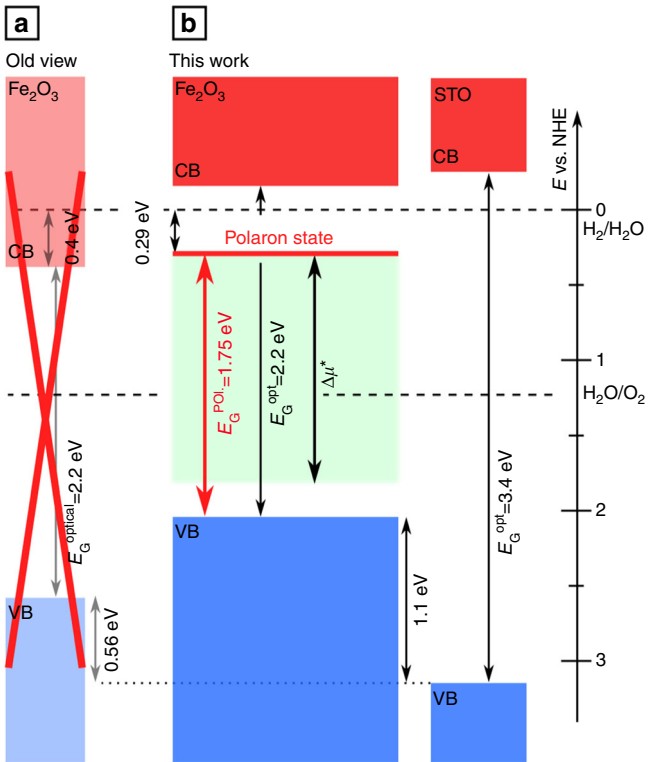

**Fig. 3** Graphical representation of the results of this work. In **a** the incorrect band edge positions of hematite as found in literature are shown[13]. **b** The band edge positions as derived in this work. The polaron level is found to be separated by the polaron gap of 1.75 eV from the valence band maximum of hematite. This level actually corresponds to the band edge, which is found to be about 0.4 eV below the hydrogen potential by electrochemical methods. Due to this finding the valence band maximum of hematite has to be raised in energy. The range of Fermi level positions from this work of $E_F - E_{VBM} = 0.25–1.8$ eV can be regarded to be the confirmed possible splitting of the quasi-Fermi level $\Delta\mu^\star$

by interface formation. In general, low and high work function contact materials should raise and lower the Fermi level at the interface, respectively. Fermi level positions differing by more than 1 eV have been achieved with Sn-doped $In_2O_3$ (ITO) and $RuO_2$ as contact materials in previous work[37]. Deposition conditions can be found in Supplementary Table 2.

In the course of an interface experiment, the contact material is deposited in a stepwise process onto the substrate. Such experiments are widely used to determine energy band alignment between materials[38]. Here we are simply interested in how far the Fermi energy can be raised or lowered during contact formation, which is extracted from the binding energy shift of the Fe2p core-level. In a previous experiment, the Fermi energy of a hematite thin film was lowered towards $E_F - E_{VBM} = 0.85$ eV by deposition of $RuO_2$[39]. We have reproduced this value using in situ prepared hematite films. Lower Fermi energy positions of $E_F - E_{VBM} = 0.4$ eV are obtained with deposition of NiO, which agrees with experiments using $SrTiO_3$ single crystals[37]. The evolution of the Fermi energy with with increasing NiO thickness is shown in Fig. 3a (blue squares and lines). A nominally undoped hematite film has been used for this experiment. Comparable behavior is observed for doped films as illustrated in Supplementary Figure 4.

Interface formation with ITO was studied using an undoped and a Zr-doped hematite film, having initial Fermi level positions of $E_F - E_{VBM} = 1.1$ and and 1.68 eV, respectively. Evaluation of the Fe2p core-level required a subtraction of the superimposed $Sn3p_{3/2}$ core-level. Details of the procedure are described in the

Supplementary Methods. The evolution of the binding energy for both experiments is included in Fig. 2. In both cases, the Fermi level increases with ITO deposition and reaches a final value of $E_F - E_{VBM} = 1.8$ eV. Figure 2b shows the $Fe2p_{3/2}$ emissions recorded before the first ITO deposition. In Fig. 2c–e the same emission is shown with a growing ITO film on the hematite substrate. The spectra recorded from the undoped sample are shown by dashed blue lines while those from the Zr-doped substrate by red solid lines. The spectra are normalized and shifted to the same binding energy for comparison of the line shape.

While the spectra of the two bare substrates in Fig. 2b exhibit no difference, the $Fe2p_{3/2}$ line of the Zr-doped sample develops a shoulder at low binding energy, as indicated by the arrow in Fig. 2c. This shoulder indicates the presence of $Fe^{2+}$[36]. It becomes more intense with increasing ITO thickness in Fig. 2d. For the undoped sample, the shoulder is only observed after the last ITO deposition step in Fig. 2e. Considering the evolution of the Fermi energy in the course of ITO deposition for the two experiments, the reduction of Fe from $Fe^{3+}$ to $Fe^{2+}$ occurs if the Fermi energy raises above $E_F - E_{VBM} = 1.75$ eV. The highest Fermi level position it found in the last deposition step shown in Fig. 2e to be 1.8 eV for both samples.

**Polaron formation as Fermi level limitation**. The consequence of this observation is that hematite $Fe_2O_3$ is not stable for Fermi energies higher than $E_F - E_{VBM} = 1.8$ eV. A small fraction of Fe atoms can change their oxidation state in the given crystallographic structure. However, raising the Fermi above $E_F - E_{VBM} = 1.8$ eV will change the oxidation state of all Fe atoms, giving rise to a huge electrostatic repulsive energy. In other words, the Fermi energy in hematite cannot be higher than $E_F - E_{VBM} = 1.8$ eV above the valence band maximum in order to limit the concentration of charged Fe. This observation provides a natural explanation for the low photovoltages observed with hematite due to those polaron trapping states related to the formation of reduced $Fe^{2+}$ if the charge carrier transport and transfer rates are slow compared to charge carrier trapping times. As a consequence the achievable photovoltages (splitting of quasi-Fermi levels) are considerably smaller than expected for a material with a bandgap of 2.2 eV[10,19].

The formation of $Fe^{2+}$ in hematite is a well-known phenomenon, described as trapping of electrons from the conduction band (polaron formation). In fact, the upper limit of the Fermi energy of $E_F - E_{VBM} = 1.8$ eV corresponds well with polaron energy levels calculated using density functional theory[26,31,33,40]. Polaron formation in hematite is also discussed as a limitation for the transport of electrons. Using transient absorption spectroscopy it was shown, for example, that self-trapping of electrons is the major trapping process in hematite, which appears within 2 ps and is therefore much faster than free electron transport or transfer[41,42]. The mobility of polarons is regarded to be much lower than for free electrons and is usually in the range of $10^{-2}$ $cm^2$ $V^{-1}$[43]. The present results suggest that polarons in addition to transport limitations also provide a fundamental limit for the bulk Fermi energy position, which also determines interfacial charge transfer as demonstrated below.

If a material only allows a certain range of Fermi energies, the concept of the energy gap has to be reconsidered. Different effective energy gaps have not been considered for inorganic solids, yet. For these materials, the optical gap is usually considered to be the same as the fundamental gap, which is the difference between itinerant completely occupied valence to empty conduction band states. This gap is usually also considered for the available range of Fermi level and quasi-Fermi level

positions, independent on possible formed defect energies with electron and hole trapping[5]. In the case of hematite, electrons excited into the conduction bands will not reside there but are trapped within 2 ps as polarons[42].

The lower limit of the Fermi energy observed in our experiments is $E_F - E_{VBM} = 0.25$ eV and was achieved by Mg-doping. The origin of the lower limit cannot be resolved as there is no evidence for spectroscopic changes. Possible origins are a deep Mg acceptor level, or the formation of hole polarons such as $Fe^{4+}$ or $O^-$. Formation of hole polarons levels are, however, still under debate in literature. Braun et al.[44] observed changes in resonant photoemission spectra and tentatively associated these with the formation of $Fe^{4+}$. Theoretically, however, it was calculated that a possible hole polaron in hematite would be positioned within the valence band[31,33]. It should be noted that a possible hole polaron would limit hole transport reactions at the hematite surface. The experimentally limited downward shift of the Fermi level either by doping or contact formation seems to indicate also an existing polaron/charge transfer defect state close to the valence band.

**The band edge positions of hematite**. The poor efficiency for hydrogen evolution has been assigned to a too low position of the $Fe_2O_3$ conduction band minimum is correct, as CBM, which is reported to be 0.4 eV below the hydrogen potential[13]. Such a level corresponds to a hematite electron affinity of 4.9 eV or ionization potential of 7.1 eV[2]. In contrast, the ionization potential measured using PES on samples without potential dopant segregation amounts to $6.6 \pm 0.1$ eV (See Supplementary Figure 9). A direct determination of the band edge energies can be obtained from interface experiments. We have used niobium-doped $SrTiO_3$ (STO) as a reference, which has been studied in the context of water splitting in the past[2,45–48]. According to the most recent literature the conduction and valence band edges of STO at pH = 0 are 0.26 eV above and 2.94 eV below the hydrogen potential, respectively[2]. These values can be translated into an electron affinity of $\chi = 4.24$ eV and an ionization potential $I_P = 7.44$ eV. The single crystalline STO sample in the present experiment has an ionization potential of 7.31 eV, in good agreement with the present determination and other literature[35].

The interface experiment, details are described in the Supplementary Discussion and Supplementary Figure 8, reveals a valence band edge of hematite at 1.1 eV above that of STO. This coincides with the band alignment derived using transitivity of interfaces with $RuO_2$ as reference material[49]. According to this alignment, the energy bands of $Fe_2O_3$ are ≈0.5 eV higher than that obtained from electrochemical measurements. The discrepancy can be explained by the $Fe^{2+}$ defect level. Band alignment in electrochemical studies is connected to charge transfer at the interface, which, for electrons from or to $Fe_2O_3$ will take place to or from the $Fe^{2+}$ level and not from the conduction band minimum is correct, as CBM. The different alignments are therefore consistently explained using the different energy gaps as illustrated in Fig. 3.

The reason why electrochemical studies have misplaced the band edges in the past can be found in their procedure. In e.g. flatband potential measurement the Fermi level of the majority charge carriers is being positioned with respect to a reference electrode and the band edge position is deduced from the determined doping concentration. For hematite the band edge that was being measured was assumed to be the conduction band minimum is correct, as CBM. Our results, however, reveal that in reality the polaron state has been measured. As a consequence, using the optical bandgap of 2.2 eV the valence band is positioned on a too low level. Our results on the band edges are direct

measurements of valence band offsets between hematite and ITO and STO. This makes the results presented here more trustworthy in aligning the band edges of different material toward each other than results from indirect approaches such as flatband potential measurements.

The polaron gap concept is not limited for hematite only but covers any other material that shows polaron formation with the polaron level within the conventional bandgap as has been theoretical shown e.g. in $BiVO_4$[50]. In addition, it is applicable also for materials that either show hole polarons such as MnO or even the formation of both, electron and hole polarons[31]. Two limiting mechanisms need to be considered: at first the energetic alignment to the contact phase e.g. the electrolyte must consider the position of the occupied (electron or hole) polaron states, which may considerably shift the effective band edge positions. In addition, for the fast charge carrier trapping in the bulk of the semiconductor, which can compete with charge transfer, the splitting of the quasi-Fermi level positions ($\Delta\mu^*$) is limited by these polaron states as well. This is depicted in Fig. 3.

Thus, for energetic converting interfaces especially in photo-electrochemical devices using oxides with localized $d^n$ electron configuration the position of the electron and hole polaron states/charge exchange states must be considered for defining the effective bandgap. We expect that similar results will be valid also for other oxides used for water splitting and must be considered for material's selection.

The future selection of materials for photocatalysis needs to take the polaron gap concept into consideration as we have conclusively shown that the present design solely based onto the optical bandgap has major short-comes that are responsible for the tremendous differences in theoretical and real performances of present devices.

## Methods

**Sample preparation**. The hematite thin films were deposited by means of RF-magnetron co-sputtering in the DArmstadt Integrated SYstem for MATerials Research. A representation of the laboratory and the co-sputtering chamber can be found in Supplementary Figure 1. Supplementary Table 1 shows the deposition conditions that were used to deposit the undoped and doped thin films.

Deposition conditions for the contact materials of the interface experiments to ITO, $RuO_2$, and NiO can be found in Supplementary Table 2. The procedure of an interface experiment is described in the Supplementary Methods at Supplementary Figure 2.

**Sample characterization**. For the acquisition of XP core-level and valence band spectra monochromatic $AlK_\alpha$ radiation ($hv = 1486.6$ eV) was used. The energy resolution has been determined from the Gaussian broadening of a sputter cleaned Ag sample to be 400 meV. The angle between detector and sample surface was 45°. Calibration of the binding energies was achieved by adjusting to the $Ag3d_{5/2}$ core-level ($E_B = 368.26$ eV) and the Ag Fermi-edge from freshly sputter cleaned samples. To avoid charging effects during the X-ray photoelectron spectroscopy (XPS) measurement platinized quartz or sapphire substrates were used for all samples.

Most XPS core-level binding energies were obtained by a fitting procedure with a Gaussian shape to the topmost part of the core-level main line. A linear extrapolation of the lower part of the low binding energy side of the valence band was used to find $E_F - E_{VBM}$. Special care is needed if the line shape of core-level or valence band changes. In these cases the fitting of the topmost part of the core-level might not be applicable. A manual procedure to find the center of the peak in the topmost part can be used instead. The valence band maximum can be extracted manually for most line shapes. Samples were a smearing out of the valence band maximum prohibited a precise determination of $E_F - E_{VBM}$ were omitted from this study. The margin of error of these procedures can be estimated to be about ±0.1.

Ultraviolet/visible/near-infrared transmission ($T$) and reflection ($R$) spectra were recorded in a range of 300–500 nm in an Agilent Cary 7000 Universal Measurement Spectrophotometer on samples deposited on bare quartz or sapphire substrates. The absorption coefficient was calculated by $\alpha = \ln((100 - R)/T)/(d/\cos(\phi_S))$, where $d$ is the film thickness and $\phi_S$ is the angle between incident beam and sample normale during the measurement. The film thickness has been obtained by measuring the edge between deposited film and substrate at different positions of the sample using a DektakXT profilometer.

XRD measurements were carried out in $\theta - 2\theta$ geometry in a Seifert XRD 30003 PTS-3 (Phase Texture Stress) using $CuK_\alpha$ radiation. Raman spectra were recorded

with the 514.5 nm emission line from an Argon multiline laser in a LabRAM High Resolution Raman/Fourier-Transform Infrared Spectroscope HR800. The measurement range was 100–1800 cm$^{-1}$. The data show the formation of well-defined hematite films in correspondents to data from literature[28].

## Data availability

The source data necessary to support the findings of this paper are available from the corresponding author upon request.

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

## Acknowledgements

The funding of this work by the European Union under the EJD-FunMat-project (Project-ID: H202-MSCA-ITN-2015) is gratefully acknowledged.

## Author contributions

C.L. conducted the experiments that were design in co-operation with A.K. The data were analyzed and interpreted by C.L. and A.K. with final additions from W.J. to the conclusion. A.K. concluded on the band alignment of hematite in presence of the polaron

state and gave the original interpretation of the data in terms of the polaron gap. C.L. and A.K. wrote the manuscript. All authors contributed to revisions.

## Additional information

**Competing interests:** The authors declare no competing interests.

