## [Peer Review File · Nature Communications]

Reviewers' comments:

Reviewer #1 (Remarks to the Author):

In this manuscript, the authors carried out the spectroscopic analysis on the hematite with various dopants, which is a popular material for solar water splitting. The authors found that the Fermi level is pinned at the energy well below the conduction band minimum and ascribe this to the formation of Fe^{2+} or electron polaron. Inspired by the observation, the authors proposed "polaron gap" in which the band gap is fundamentally limited by the self-trapped polarons at the lattice site. The experiment was carried out systematically and results are convincing. However, the electron polaron in hematite is already well known in literature, as correctly referenced in the manuscript, and the effective Fermi level pinning due to the self-trapped polaron is also widely assumed in the transition metal oxides although it may not have been explicitly emphasized. As such, the proposal of "polaron gap" is hardly new and just a redefinition of what is already well established. (In fact, the word "polaron gap state" is frequently used in organic semiconductors.) In this respect, I do not think this work carries an impact as required in Nature Communications. However, the experimental part is worth publication in a more specialized journal.

Reviewer #2 (Remarks to the Author):

The paper by Jaegermann and co-workers reports on the importance of considering polaron formation as the effective doping limit for operative "band edge" energies that define the redox potentials in transition metal oxides like hematite. The main message is that the classical band edge positions in hematite have been mislocated, because the operative ones involved in water splitting are polaron-based intra-gap states. The study starts with some basic semiconductor concepts about self-compensation to build its arguments about $\text{Fe}^{2+}/\text{Fe}^{3+}$ valence states in hematite, and from there relies mostly on measured photoelectron and photoabsorption spectra on n- and p-type doped hematite thin films to "prove" its arguments.

The study is fairly soundly formulated and well written and illustrated. The arguments are generally backed up by the presented data. I found the study interesting and enjoyable to read. And I do think the contribution is significant and timely enough to warrant consideration by a higher impact journal with a broad-based readership like Nature Communications. However, I'm not yet convinced that the claim should go so far to be presented as "A new band gap concept" as if conceived entirely ab initio – this might need more context. In addition to elaborating on this point, below I mention some aspects of the story that could be strengthened, before the manuscript should move further along towards publication in Nat Comm.

1) Is the polaron gap an entirely new concept (for transition metal oxides), or is it just one that is not often considered explicitly in current mainstream photoelectrochemistry? As I'm sure the authors must be aware, there are many books on polarons and polaronic semiconducting materials from the early days of solid-state physics – that charge transport behavior (mobility, conductivity, temperature-dependence) can be defined by polarons instead of itinerant band carriers is well known. The present paper appears to be making the point that the energy levels for polaronic states make band edges like the CBM of hematite irrelevant, in the context of water splitting. I believe this to be a mix of old and new concepts, but without clarification not entirely new. This is evidenced by the fact that there have been many studies whose entire goal has been to locate the energy level of localized "defect" states like polarons with the classical band gap, probing them with photoelectron and inverse photoemission spectroscopies, because of its important contribution to reactive behavior (particularly redox behavior at interfaces). The new part that I think the present paper can indeed lay claim to is that if indeed the CBM water reduction potential has been 'hijacked' by the polaronic states ~ 0.4 eV lower in energy then this indeed should lead to a much less efficient HER. I believe the authors should take this rather general set of comments about 'the title' and main messaging to heart and provide a revised version in which the claims of significance have been better contextualized, so as to avoid unnecessary inflation.

2) Explicit attention should be paid to questions running through the reviewer's minds about the

role of polaronic mobility, along the following lines:

a. Polarons are generally considered much less mobile than electrons in itinerant states. They would also tend to be very dilute, as the authors correctly point out. Yet polaronic mobilities have not been discussed, only their rates of formation. The early ab initio work of Rosso and co-workers is particularly relevant.

b. Even though they are lower in energy and thus typically occupied by fast thermalization of photoexcited electrons does not necessarily mean polarons are more mobile than the fraction of CBM carriers still present. Can the authors do a 'mass balance' calculation to estimate the relative concentrations of polaronic versus thermal carriers at the CBM, and then on the basis of their respective mobilities estimate the relative conductivities of the two populations? This would help cement the argument that polarons are both energetically relevant and of sufficient mobility.

c. Hole polarons (VBM) are also mentioned but dismissed as debatable in terms of their host sublattice. I think more attention should be paid here, again on the notion of mobilities, because in the context of water splitting separation of charge at the oxide/electrolyte interface is essential. The current paradigm with respect to transport (as per the above) is that the hole polaron is rate limiting – if the hole polaron controls the effectiveness of charge separation then its mobility is relevant to the main argument of the paper.

3) One of the main conclusions of the paper is that the VBM is actually more than 0.5 eV higher in energy than previously thought. This is a substantial difference that is given little attention. Although I do not think it warrants additional experiments to be performed, I do think it warrants a bit more explanation and connection to previous work. This could be in the form of elaborating about how such a difference is conceptually easily missed (say in electrochemical measurements of the OER photo-onset), or perhaps previous literature provides sufficient proof of this prospect, or its ambiguity. Some articulation about how such a big difference could have persisted overlooked even to this day would also go a long way in justifying the general significance of novelty of the study for high impact publication.

In summary, I am supportive of eventual publication of the paper in Nat Comm, if the authors can strengthen the scholarly arguments and presentation satisfactorily along these lines.

Reviewer #3 (Remarks to the Author):

In this manuscript, the Authors use photoelectron spectroscopy to measure various doped and interfaced iron oxide films. From these measurements, they conclude that polaron formation can place a limit on the Fermi level energy, which in turn explains the low photovoltages and measured efficiencies for iron oxide photoelectrodes. Overall, this reviewer finds the manuscript sound and well written. Given that metal oxides are still being routinely investigated for photoelectrochemistry, despite consistently low performances, I also believe this manuscript has the broad impact needed for publication in Nature Communications. In general, the effects of polarons in photocatalysis seem to be coming to center stage, and this is another vital piece of evidence when considering their use.

Before publication, the following minor points should be addressed:

1. The Authors should detail a little more about how the photoemission data was analyzed to get the peak energy and offset shifts. Was the slope extrapolated or just the intercept used? How does this work with a change in shape or linewidth of the VB photoemission? What are the error bars on these numbers?

2. It would be excellent if the Figure 2 middle panel could be separated or clarified. There is a lot of information going on in the middle panel and it is slightly difficult to interpret, but this is the main data of the paper. Maybe the authors could separate the doping and interface studies in to separate figures?

3. With the different doping and interface studies, was there any significant changes in the optical spectrum as well? As in, the fermi-level to valence band offset is discussed, but what about the valence band to conduction band offset. Although this can be complicated to determine exactly, it may be worth commenting on further.

We thank the reviewers for their opinion and suggestions to improve our manuscript. In the following we have answered to their statements point-by-point.

Reviewer #1 (Remarks to the Author):

In this manuscript, the authors carried out the spectroscopic analysis on the hematite with various dopants, which is a popular material for solar water splitting. The authors found that the Fermi level is pinned at the energy well below the conduction band minimum and ascribe this to the formation of Fe²⁺ or electron polaron. Inspired by the observation, the authors proposed "polaron gap" in which the band gap is fundamentally limited by the self-trapped polarons at the lattice site. The experiment was carried out systematically and results are convincing. However, the electron polaron in hematite is already well known in literature, as correctly referenced in the manuscript, and the effective Fermi level pinning due to the self-trapped polaron is also widely assumed in the transition metal oxides although it may not have been explicitly emphasized. As such, the proposal of "polaron gap" is hardly new and just a redefinition of what is already well established. (In fact, the word "polaron gap state" is frequently used in organic semiconductors.) In this respect, I do not think this work carries an impact as required in Nature Communications. However, the experimental part is worth publication in a more specialized journal.

Answer:

A statement was added which acknowledges the reviewers remark on the organic semiconductors. In addition, the title of the manuscript was changed. It should be stated, that the investigated effects of the "polaron gap state" reducing the effective photovoltage have not yet received the attention for the application of hematite as photoelectrode. Our interpretation suggests that similar band gaps given by the polaron states are operative in many oxide photoelectrodes. Hence, in this context these are not merely states within the band gap but form the band edges for charge transfer reactions. To the best of our knowledge this interpretation has not yet been discussed for transition metal oxides in this depth.

Reviewer #2 (Remarks to the Author):

The paper by Jaegermann and co-workers reports on the importance of considering polaron formation as the effective doping limit for operative "band edge" energies that define the redox potentials in transition metal oxides like hematite. The main message is that the classical band edge positions in hematite have been mislocated, because the operative ones involved in water splitting are polaron-based intra-gap states. The study starts with some basic semiconductor concepts about self-compensation to build its arguments about Fe²⁺/Fe³⁺ valence states in hematite, and from there relies mostly on measured photoelectron and photoabsorption spectra on n- and p-type doped

hematite thin films to “prove” its arguments.

The study is fairly soundly formulated and well written and illustrated. The arguments are generally backed up by the presented data. I found the study interesting and enjoyable to read. And I do think the contribution is significant and timely enough to warrant consideration by a higher impact journal with a broad-based readership like Nature Communications. However, I’m not yet convinced that the claim should go so far to be presented as “A new band gap concept” as if conceived entirely ab initio – this might need more context. In addition to elaborating on this point, below I mention some aspects of the story that could be strengthened, before the manuscript should move further along towards publication in Nat Comm.

1) Is the polaron gap an entirely new concept (for transition metal oxides), or is it just one that is not often considered explicitly in current mainstream photoelectrochemistry? As I’m sure the authors must be aware, there are many books on polarons and polaronic semiconducting materials from the early days of solid-state physics – that charge transport behavior (mobility, conductivity, temperature-dependence) can be defined by polarons instead of itinerant band carriers is well known. The present paper appears to be making the point that the energy levels for polaronic states make band edges like the CBM of hematite irrelevant, in the context of water splitting. I believe this to be a mix of old and new concepts, but without clarification not entirely new. This is evidenced by the fact that there have been many studies whose entire goal has been to locate the energy level of localized “defect” states like polarons with the classical band gap, probing them with photoelectron and inverse photoemission spectroscopies, because of its important contribution to reactive behavior (particularly redox behavior at interfaces). The new part that I think the present paper can indeed lay claim to is that if indeed the CBM water reduction potential has been ‘hijacked’ by the polaronic states ~ 0.4 eV lower in energy then this indeed should lead to a much less efficient HER. I believe the authors should take this rather general set of comments about ‘the title’ and main messaging to heart and provide a revised version in which the claims of significance have been better contextualized, so as to avoid unnecessary inflation.

Answer:

The title has been . The authors acknowledge that previously polarons/charge transitions have been already investigated (e.g. redox behavior at interfaces).

The reviewer shares the opinion of the authors that the “hijacking” of the CBM by the polaronic state has not yet been discussed. We have added a statement on polarons in organic and oxide semiconductors. In addition, statements which might be misinterpreted as if the authors claimed to be the first to discuss charge transition involving polaron states have been changed in order to acknowledge previously conducted studies.

It has to be stated, however, that our experimental approach provides a rather direct way to assign the low photovoltages directly to the formation of polarons and their appearance in the optical band gap.

2) Explicit attention should be paid to questions running through the reviewer’s minds about the role of polaronic mobility, along the following lines:

a. Polarons are generally considered much less mobile than electrons in itinerant states. They would also tend to be very dilute, as the authors correctly point out. Yet polaronic mobilities have not been

discussed, only their rates of formation. The early ab initio work of Rosso and co-workers is particularly relevant.

Answer:

The reviewer is right in the claim that mobilities of polarons have been neglected in the first manuscript. We have added a statement and followed the reviewers suggestion to cite Rosso and co-workers in this aspect.

b. Even though they are lower in energy and thus typically occupied by fast thermalization of photoexcited electrons does not necessarily mean polarons are more mobile than the fraction of CBM carriers still present. Can the authors do a 'mass balance' calculation to estimate the relative concentrations of polaronic versus thermal carriers at the CBM, and then on the basis of their respective mobilities estimate the relative conductivities of the two populations? This would help cement the argument that polarons are both energetically relevant and of sufficient mobility.

Answer:

This question is of great importance indeed. It is, however, an issue that concerns the kinetic aspects of polaron formation on the performance of hematite and TMOs in general. Our study is focused on the energetic position of polarons and their impact on the photovoltages. The suggested calculations would be of interest to the impact of polaron formation on the photocurrent. It is our believe that such a study is very worthy to be conducted as a independent work in the future combining previously reported results on the formation rate and mobility with our work on the position of the polarons.

c. Hole polarons (VBM) are also mentioned but dismissed as debatable in terms of their host sublattice. I think more attention should be paid here, again on the notion of mobilities, because in the context of water splitting separation of charge at the oxide/electrolyte interface is essential. The current paradigm with respect to transport (as per the above) is that the hole polaron is rate limiting – if the hole polaron controls the effectiveness of charge separation then its mobility is relevant to the main argument of the paper.

Answer:

The reviewer is completely right in his statement on the importance of the hole polaron with respect to rate limitation. The spectroscopic results in our studies on the hole polaron are, however, not as clear as for the electron polaron. In the previous version of the manuscript we had already discussed this point.

Now, we have added a statement and acknowledged the reviewer remark that the hole polaron controls the charge separation. We believe it would be worthy to conduct further dedicated studies in order to identify the exact nature (and position) of the hole polaron in hematite and TMOs in general.

3) One of the main conclusions of the paper is that the VBM is actually more than 0.5 eV higher in

energy than previously thought. This is a substantial difference that is given little attention. Although I do not think it warrants additional experiments to be performed, I do think it warrants a bit more explanation and connection to previous work. This could be in the form of elaborating about how such a difference is conceptually easily missed (say in electrochemical measurements of the OER photo-onset), or perhaps previous literature provides sufficient proof of this prospect, or its ambiguity. Some articulation about how such a big difference could have persisted overlooked even to this day would also go a long way in justifying the general significance of novelty of the study for high impact publication.

Answer:

We are very grateful for the reviewers opinion on the importance of this aspect of our study. We followed his suggestion and added statements on the previous works and how the position of the valence band could have been missed. In addition, we decided to slightly change Figure 3 to show the difference in band position even clearer.

In summary, I am supportive of eventual publication of the paper in Nat Comm, if the authors can strengthen the scholarly arguments and presentation satisfactorily along these lines.

Reviewer #3 (Remarks to the Author):

In this manuscript, the Authors use photoelectron spectroscopy to measure various doped and interfaced iron oxide films. From these measurements, they conclude that polaron formation can place a limit on the Fermi level energy, which in turn explains the low photovoltages and measured efficiencies for iron oxide photoelectrodes. Overall, this reviewer finds the manuscript sound and well written. Given that metal oxides are still being routinely investigated for photoelectrochemistry, despite consistently low performances, I also believe this manuscript has the broad impact needed for publication in Nature Communications. In general, the effects of polarons in photocatalysis seem to be coming to center stage, and this is another vital piece of evidence when considering their use. Before publication, the following minor points should be addressed:

1. The Authors should detail a little more about how the photoemission data was analyzed to get the peak energy and offset shifts. Was the slope extrapolated or just the intercept used? How does this work with a change in shape or linewidth of the VB photoemission? What are the error bars on these numbers?

Answer:

A statement on the details of data analyzation and error of the method has been added to the experimental section of the manuscript.

2. It would be excellent if the Figure 2 middle panel could be separated or clarified. There is a lot of information going on in the middle panel and it is slightly difficult to interpret, but this is the main data of the paper. Maybe the authors could separate the doping and interface studies in to separate figures?

Answer:

The separation of the middle panel in Figure 2 would change the logical framework of the manuscript. We have decided to leave it as it is but added a remark that Figure S7 shows the same data with more information given. Here, we have added an explanation to the figures subtext.

3. With the different doping and interface studies, was there any significant changes in the optical spectrum as well? As in, the fermi-level to valence band offset is discussed, but what about the valence band to conduction band offset. Although this can be complicated to determine exactly, it may be worth commenting on further.

Answer:

We have added representative optical spectra to the Supplementary Information and added a statement in the text. There are no significant changes to the optical spectrum. It can be concluded that the optical band gap is conserved.

REVIEWERS' COMMENTS:

Reviewer #1 (Remarks to the Author):

Even though authors responded to my criticism, I still maintain the original opinion of rejection. The fact that polaronic states can limit the usable band gap of doped oxides is conceptually so obvious. If it has not been appreciated explicitly in applying the hematite to the photoelectrode, it should be because the technical issue is focused on improving the carrier mobility, rather than tuning the band gap. This in turn means that the impact of the present work is not high enough for the publication in Nature Communications.

Reviewer #2 (Remarks to the Author):

The authors have done a satisfactory job of addressing my original comments in the revised manuscript. I fully support publication in Nat Comm as is.

Reviewer #3 (Remarks to the Author):

This Reviewer's comments are mostly addressed. For the added Figure S7, it is still quite cluttered, and I hope the authors will at least describe the different panels in the caption. If the other Reviewers are satisfied that the paper has been properly placed within the larger context of the field of polarons, I am okay with publication. I agree, however, that this is a concern which must be addressed.

The reviewers' comments and our response to these points are shown below:

Reviewer #1 (Remarks to the Author):

Even though authors responded to my criticism, I still maintain the original opinion of rejection. The fact that polaronic states can limit the usable band gap of doped oxides is conceptually so obvious. If it has not been appreciated explicitly in applying the hematite to the photoelectrode, it should be because the technical issue is focused on improving the carrier mobility, rather than tuning the band gap. This in turn means that the impact of the present work is not high enough for the publication in Nature Communications.

Answer:

We do not share the opinion of the reviewer that the limitation of the usable band gap in inorganic solids is thoroughly accepted in the scientific community. As is correctly stated for hematite the main focus right now is the improvement of the carrier mobility. This, however, ignores the fact that one additional important prerequisite for the usage of hematite as a photoelectrode related to the energy states involved in photoreactions (defining the photopotential) is just not met. This is shown by our study. We strongly believe that the limitation of the band gap by polaron states has to be considered in the consideration of new materials for the use as photoelectrodes.

Reviewer #2 (Remarks to the Author):

The authors have done a satisfactory job of addressing my original comments in the revised manuscript. I fully support publication in Nat Comm as is.

Answer:

We thank the reviewer for the original comments and the support of our manuscript.

Reviewer #3 (Remarks to the Author):

This Reviewer's comments are mostly addressed. For the added Figure S7, it is still quite cluttered, and I hope the authors will at least describe the different panels in the caption. If the other Reviewers are satisfied that the paper has been properly placed within the larger context of the field of polarons, I am okay with publication. I agree, however, that this is a concern which must be addressed.

Answer:

We thank the reviewer for the comments and the support of our manuscript. We have added explanations to the different panels of the respective Figure (now Supplementary Figure 9) in the caption.